# Coupling analysis of risk factors in road cargo transport accidents and preventive measures with an N–K model

Runhua Huang[1,2]*, Huichao Guo[2]

1 School of Management and Economics, The Chinese University of Hong Kong, Shenzhen, Guangdong, China, 2 School of Economics and Management, Zhejiang Business College, Hangzhou, Zhejiang, China

* runhuahuang@link.cuhk.edu.cn

**Data availability statement:** The data has been uploaded to Github and you can access that with the link: https://github.com/RunhuaHuang/Data-Pone.

## Abstract

Road cargo transport accidents have been rising in China, posing threats to economic stability and public safety. This study identifies key risk factors in road freight accidents by analysing 160 accident investigation reports and employs a combined N–K model and complex-network approach to evaluate coupling effects among factors. Twenty-seven specific risk factors were classified under individual, vehicle, environment and management domains. Multi-factor coupling (e.g. I–V–E–M) markedly increases accident probability, and complex-network metrics highlight hub (overloading, brake defects, inadequate road inspection, failure to observe road conditions) and bridge (inadequate inspection, illegal lane change, brake defects, speeding) factors. Targeted enforcement of loading limits, enhanced inspection regimes and driver training are recommended. Integrating the N–K model with complex-network analysis provides a systematic basis for prioritising preventive measures and improving road-freight safety.

## Introduction

Road cargo transport refers to the process of moving goods on roads using specific vehicles, typically trucks. With the robust growth of China's economy, continuous industrial development, and expansion of newly constructed highways, road cargo transport has become an increasingly significant economic activity and a key component of China's comprehensive transportation system [1]. However, relatively low entry barriers in the trucking industry, combined with a vast number of operators and diverse types of cargo, have made it challenging to maintain consistent safety standards [2,3].

Official data show that although freight trucks constitute less than 10% of all motor vehicles in China, they are responsible for a disproportionately high percentage (around 25%) of total traffic accidents [4]. Such accidents not only threaten public safety but also pose serious risks to economic stability, particularly because accidents involving trucks carrying hazardous materials or high-value goods can result in large-scale casualties and financial losses [5,6]. Consequently, improving safety in road freight transport has become a critical policy concern.

**Funding:** This research was funded by the Zhejiang Provincial Department of Culture, Radio, Television and Tourism, grant number 2024KYZ009. The funders had no role in study design, data collection and analysis, decision to publish, or preparation of the manuscript.

Existing research has sought to identify major risk factors and propose preventive measures for road transport. Techniques include Bayesian Networks (BNs) for causation analysis [2,7], integrated risk management frameworks [8], and route optimization models that minimize both cost and accident risk [9]. While Bayesian Networks are useful for mapping probabilistic dependencies among factors, they often require extensive prior data and can be less intuitive for directly quantifying complex multi-factor interactions or "coupling." In contrast, the N-K model provides a more direct way to represent and measure how sets of interrelated factors (each with multiple possible states) influence the overall "fitness" (or in this case, accident likelihood) of the system [4,10,11]. By varying the K parameter, one can analyze how changes in the connectivity among factors (such as individual behavior, vehicle conditions, environmental hazards, and management practices) affect the emergent risk level.

Nevertheless, the N-K model alone is not sufficient to capture the network structure of risk factor interactions in fine detail. Complex network theory helps visualize and quantify relationships among numerous factors by examining node centralities and network topologies [12–14]. Integration of these two methods can thus yield a comprehensive approach to analyzing both the probabilistic coupling of factors (via the N-K model) and the topological significance of particular factors (via network centralities). This combined method can help policymakers and industry stakeholders focus on the most critical risk elements and their interdependencies, thereby guiding more targeted interventions [15,16].

In the present study, we collected and analyzed over 160 accident investigation reports from various regions in China. We extracted specific risk factors documented as either direct or major contributory causes, then used the "4M" framework—individual, vehicle, environment, and management—to classify these factors systematically. The N-K model was applied to determine how coupling (particularly multi-factor coupling) escalates accident probability, while a complex network analysis was conducted to identify key nodes via degree, closeness, and betweenness centralities. In doing so, we aim to provide a more accurate depiction of how multiple risk factors combine to produce serious accidents, and to propose countermeasures that directly address the most influential factors and couplings.

The remainder of this article is organised as follows. The Materials and methods section first details the procedure for risk-factor identification from accident reports, then describes the classification of coupling types, the construction of the N–K model, and the development of the complex-network framework. The Results section presents the quantitative outcomes of the N–K model and the network centrality analysis. The Discussion interprets these findings, including the revised weighting of factors, proposed preventive measures, and study limitations with avenues for future work. Finally, the Conclusions section summarises the main contributions and practical implications of the study.

## Materials and methods

### Risk-factor identification

Risk factor identification is the systematic and continuous process of recognizing potential causes leading to accidents before they occur [8,17]. In this study, we primarily relied on more than 200 official accident investigation reports collected from transportation authorities across different provinces in China. Each of these investigation reports typically includes root cause analysis as well as contributory and secondary factors. To extract risk factors from the reports, we performed the following steps:

1. **Initial Review:** Each accident investigation report was read in full to identify any mention of causal factors. This encompassed explicit "root causes," "direct causes,"

and "contributory factors." Factors described with synonymous terms (e.g., "excessive speed" vs. "overspeeding") were consolidated under a single representative label (e.g., "speeding").

2. **Coding and Consolidation:** We coded each reported cause or factor (e.g., "driver speeding," "vehicle brake failure," "poor road design") and consolidated synonymous or repeated items. For instance, "excessive speed" and "overspeeding" were grouped together as "speeding."

3. **Local Consultation and Confirmation:** For each accident, we attempted to verify our coding with the local traffic police who had overseen the investigation. In 160 of the 203 reports, the police confirmed or clarified our factor classifications. The remaining 43 reports were excluded from subsequent analyses due to disagreements about cause attribution, difficulty contacting the relevant officers, or insufficient cooperation. Additionally, factors that were rarely reported or considered tangential by both the investigation reports and the police were merged with more prevalent factors or removed entirely.

4. **Categorization:** We then mapped the finalized factors to four top-level categories based on the "I–V–E–M" framework: *individual*, *vehicle*, *environment*, and *management*.

After discarding infrequent items and merging closely related ones, we obtained 27 specific risk factors, as listed in Table 1. These are grouped under the four major dimensions—*individual (I)*, *vehicle (V)*, *environment (E)*, and *management (M)*—and serve as the basis for subsequent analysis. In the next section, we introduce the concept of risk-factor coupling to illustrate how these factors can interact or "couple," further heightening the likelihood of accidents.

This study used anonymized data from the official road freight transport accident investigation reports issued by the Chinese Transportation Management Department. The only human-related information in these reports is the anonymous statements provided to traffic police after the accidents, which do not include any personal or biometric data.

## Meaning and classification of risk-factor coupling

The term "coupling" originates from physics, describing mutual interactions and influences among multiple systems or modes of operation. It has been applied in diverse fields such as safety science, sociology, and economics [1,2,5]. In safety studies, coupling refers to phenomena arising between risk or causal factors [10,11,18]. Specifically, risk-factor coupling describes interdependent, mutually influencing relationships among various elements of a system. Strong interdependencies among risk factors lead to a more intense coupling effect, whereas weaker dependencies result in a diminished coupling effect [18]. In road cargo transport, the coupling of two or more risk factors can increase the probability of an accident. When two factors combine, they may create a localized coupled risk that intensifies the potential for accidents. If additional factors also couple with this initial combination, the accident probability can further escalate [15,16]. Coupling forms can be categorized by the number of participating risk factors [3–5]:

- *Single-factor coupling*: Only one factor (e.g., a defect in the vehicle's braking system) causes an accident. Although termed "coupling," this scenario essentially reflects a single point of failure without interaction with other risk elements.
- *Two-factor coupling*: Two distinct factors simultaneously contribute to an accident (e.g., overloading plus speeding).

**Table 1. Dataset of accident risk factors.**

| Primary Factor | Code | Risk Factor Description |
|---|---|---|
| Individual (I) | I01 | Speeding |
| | I02 | Fatigue driving |
| | I03 | Lane occupation |
| | I04 | Failure to observe road conditions |
| | I05 | Failure to yield |
| | I06 | Improper driving operations |
| | I07 | Unlicensed driving (no valid license) |
| | I08 | Not maintaining a safe distance |
| | I09 | Illegal lane change |
| Vehicle (V) | V01 | Illegal modification |
| | V02 | Defective braking system |
| | V03 | Defective steering system |
| | V04 | Defective tires |
| | V05 | Defective lighting system |
| | V06 | Defective safety protection devices |
| | V07 | Defective reflective marking |
| | V08 | Overloading/Oversized loading |
| Environment (E) | E01 | Adverse weather (rain, fog, etc.) |
| | E02 | Roadwork section |
| | E03 | Ramps |
| | E04 | Curved ramps |
| | E05 | Intersections |
| | E06 | Slippery road surface |
| | E07 | Incomplete traffic signs/markings |
| | E08 | Improper road design |
| Management (M) | M01 | Inadequate vehicle dynamic monitoring |
| | M02 | Inadequate road safety inspection and control |

- *Multiple-factor coupling*: Three or more risk factors act together, creating a more complex interaction scenario. For instance, a fatigued driver operating an overloaded truck on a slippery road represents a triple-factor coupling.

To better analyze these coupling effects, we incorporate a complex network approach to complement the N-K model, which by itself has limited interpretability for multi-factor interactions [12–14]. Fig 1 illustrates a conceptual risk coupling model for road cargo transport accidents, showing how various risk factors from the four dimensions (Individuals, Vehicle, Environment, Management) can interweave and heighten the likelihood of an accident.

## N–K model construction

The N-K model was initially used to solve gene combination problems [10], analyzing how interactions among factors affect changes in the entire system. Let $N$ represent the number of elements (factors) in the system; if each factor has $S$ possible states, then the total number of possible system states is $S^N$. The parameter $K$ denotes the number of interconnections among these elements (with $0 \le K \le N - 1$). When $K = 0$, none of the factors are coupled (i.e., they operate independently). When $K = N - 1$, each factor is influenced by all other factors. In essence, the overall change in the system is driven by both the coupling within each subsystem and the coupling interactions across different subsystems [2,11,19]. Suppose $a,b,c,d$ denote the four top-level risk factors: Individuals (I), vehicle (V), environment (E), and management (M). Define $h,i,j,k \in \{0,1\}$ as the states (safe = 0, unsafe = 1) of these factors. Let $p_{hijk}$ be the joint probability that the Individuals factor is in state $h$, the vehicle factor in state $i$, the environment in state $j$, and the management factor in state $k$. Meanwhile, $p_{h\cdots}$, $p_{\cdot i\cdots}$, $p_{\cdot\cdot j\cdots}$,

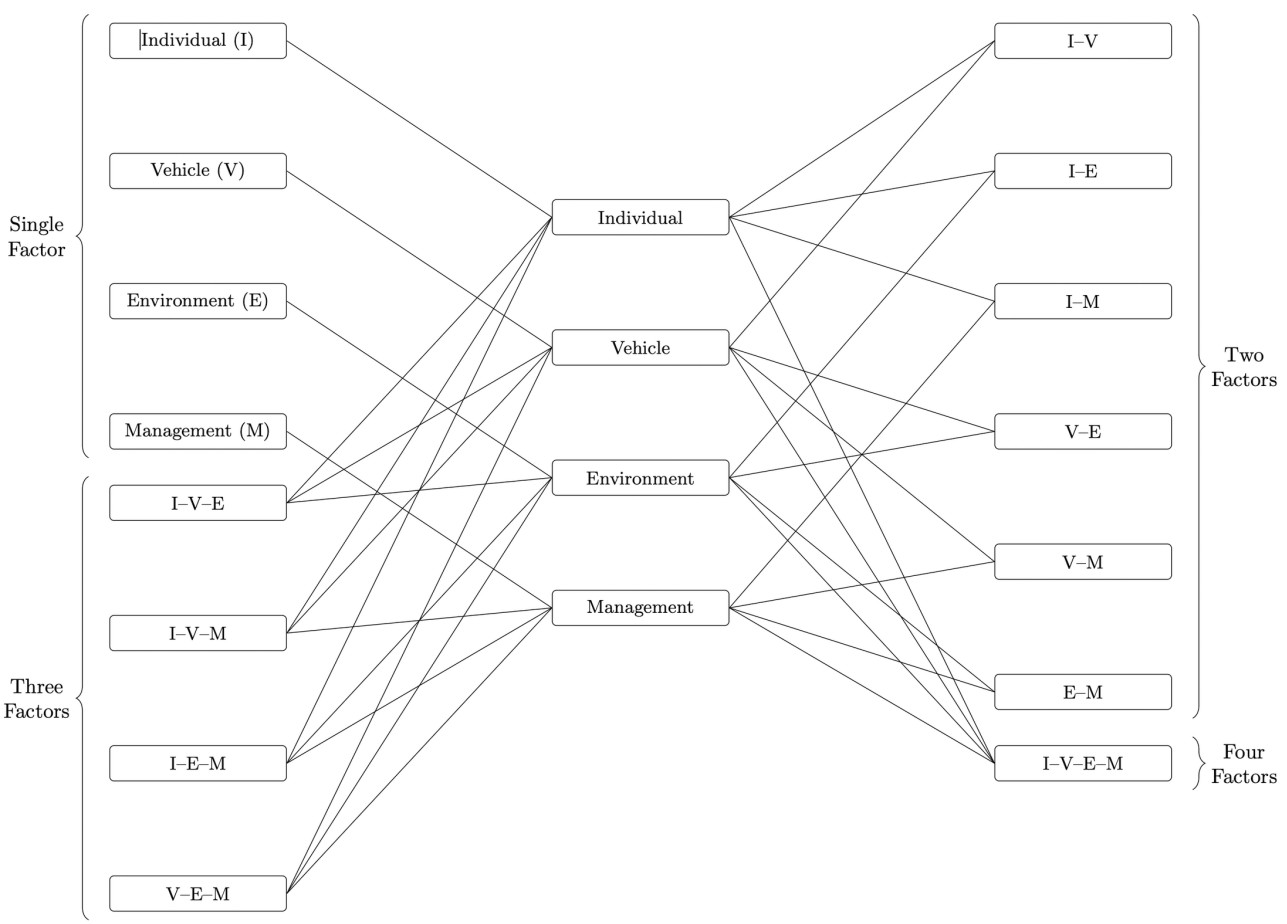

**Fig 1. Conceptual risk coupling model of road cargo transport accidents.**

and $p_{\cdots k}$ denote marginal probabilities for each individual domain. The overall risk coupling value $T(a,b,c,d)$ can be computed by

$$T(a, b, c, d) = \sum_{h=0}^{1} \sum_{i=0}^{1} \sum_{j=0}^{1} \sum_{k=0}^{1} p_{hijk} \, \log_2\!\left( \frac{p_{hijk}}{p_{h\cdots}\, p_{\cdot i\cdots}\, p_{\cdot\cdot j\cdots}\, p_{\cdots k}} \right), \tag{1}$$

where $\log_2(\cdot)$ denotes the logarithm base 2. A larger value of $T(a,b,c,d)$ indicates a higher probability of multi-factor coupling leading to an accident [16,20]. When precisely *two* categories of risk factors (e.g., Individuals and vehicle) are both in the unsafe state, the two-factor coupling value can be calculated. For instance, for Individuals–vehicle $(a,b)$ coupling,

$$T_{21}(a, b) = \sum_{h=1}^{1} \sum_{i=1}^{1} p_{hi\cdots} \, \log_2\!\big( p_{hi\cdots} \,/\, \big[ p_{h\cdots}\, p_{\cdot i\cdots} \big] \big). \tag{2}$$

Similarly, when *three* categories of factors are in the unsafe state, such as Individuals–vehicle–environment (*a*,*b*,*c*), the triple-factor coupling value is

$$T_{31}(a,b,c) = \sum_{h=1}^{1}\sum_{i=1}^{1}\sum_{j=1}^{1} p_{hij\cdots} \log_2\left(p_{hij\cdots} / \left[p_{h\cdots}\, p_{\cdot i\cdots}\, p_{\cdot\cdot j\cdots}\right]\right). \tag{3}$$

In this manner, the N-K model can effectively quantify the coupling effects among risk factors, reducing subjectivity in the evaluation. However, because it focuses on broad‑level coupling among the four top‑level categories (I, V, E, M), it does not by itself reveal finer relationships among the 27 specific risk factors or guarantee effective risk control. Consequently, we introduce a complex network approach to address this limited interpretability [15,21,22], using the risk value *T* from the N-K model to refine the network analysis and identify the critical risk factors in road cargo transport accidents.

## Complex-network construction

A complex network consists of nodes and edges, and it often exhibits properties such as self-organization, self-similarity, the existence of attractors, small-world characteristics, and scale-free degree distributions [12]. Such networks provide a powerful way to analyze systems where numerous elements and their interrelationships form a complex web [12–14]. In traffic safety research, complex networks can vividly illustrate how different risk factors interconnect and influence one another [23–26].

In this study, we construct an undirected network of road cargo transport accident risk factors. We consider the 27 identified risk factors as the network's nodes, and we draw an edge between two nodes if the corresponding factors co-occur in the same accident (based on co-occurrence analysis of the accident reports). The co-occurrence rate between any two factors describes the strength of their association. We used Gephi software to visualize this network (see Fig 2). In analyzing the risk-factor network, we focus on three centrality measures of each node: degree centrality, closeness centrality, and betweenness centrality.

**Degree centrality.**  In a network, the degree of a given node *i*, usually denoted $k_i$, is the number of other nodes directly connected to *i* (i.e., the number of edges incident on node *i*). Node degree is one of the basic characteristics of a network and reflects the position or influence of a node in the entire system. A larger degree value means the node directly connects to more nodes, implying that the node is more influential or critical in the network [12].

**Closeness centrality.**  Closeness centrality measures the proximity of node *i* to all other nodes. It is defined as the reciprocal of the sum of the shortest path distances from node *i* to all other nodes in the network [11]. In other words, the smaller the total distance from *i* to all other nodes, the higher the closeness centrality of node *i*. Geometrically, a node with high closeness is near the center of the network. The closeness centrality of node *i* can be calculated as:

$$C_i = \frac{1}{\sum_j d(i,j)}, \tag{4}$$

where $d(i,j)$ is the length of the shortest path between nodes *i* and *j*.

**Betweenness centrality.**  Betweenness centrality is a measure of a node's ability to control or mediate the transmission of information (or risk) through the network. The core idea is that if a node lies on a large number of shortest paths between other nodes, it acts as a key bridge in the network and thus has a high betweenness centrality. Formally, the betweenness

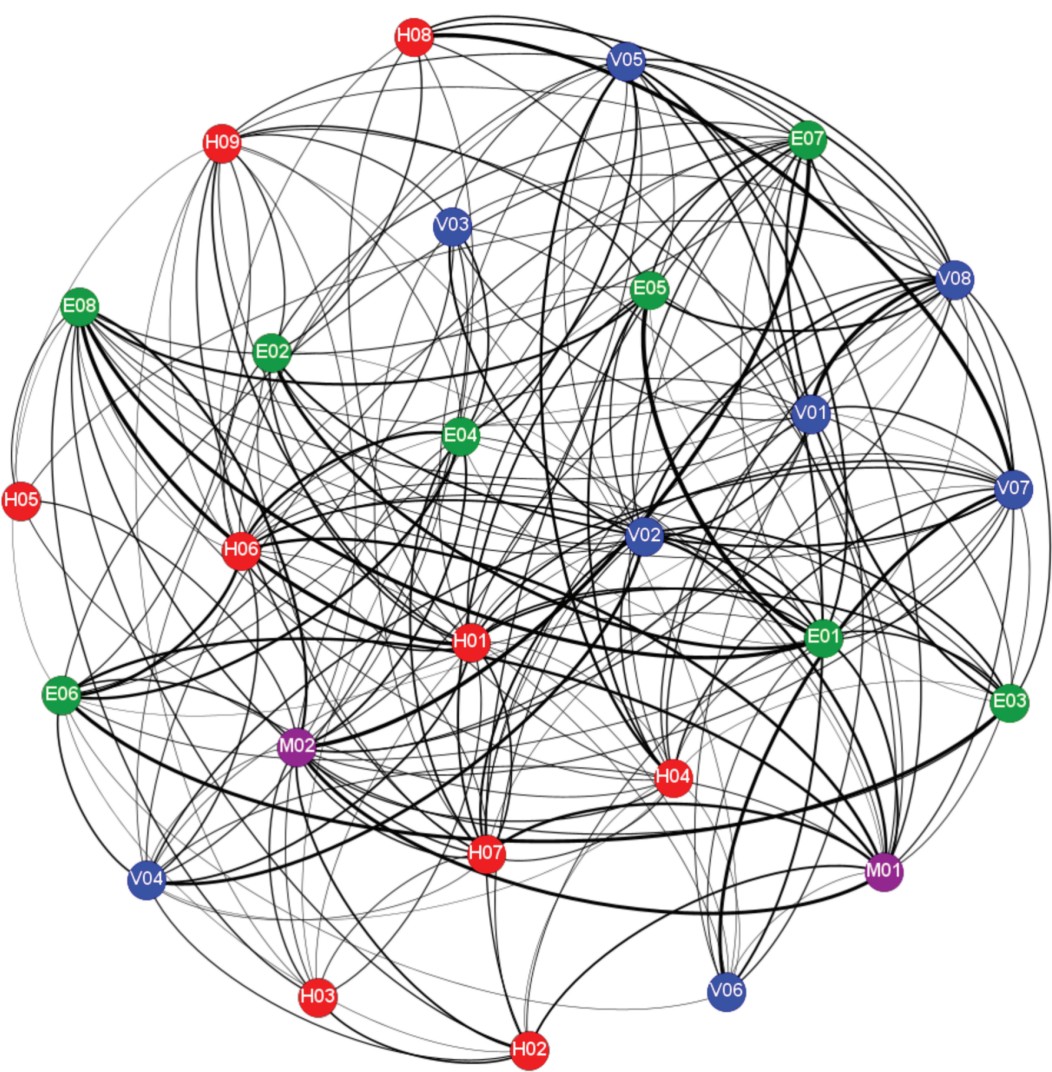

**Fig 2. Association network of risk factors in road cargo transport accidents.** Each node represents a specific risk factor (see Table 1); edges indicate co-occurrence relationships between risk factors in accident cases.

of node $v$ can be expressed as:

$$B_v = \sum_{j \neq k} \frac{g_{jk}(v)}{g_{jk}},$$

(5)

where $g_{jk}$ is the total number of shortest paths between nodes $j$ and $k$, and $g_{jk}(v)$ is the number of those shortest paths that pass through node $v$. A higher $B_v$ indicates a stronger ability of node $v$ to control risk transmission. Such bridge nodes are crucial in the network and warrant special attention [14].

## Results

### N–K model outcomes

The primary risk factors in road freight transport accidents generally include Individuals, vehicle, environment, and management. Each factor has two states, 0 and 1, representing safe

and unsafe conditions, respectively. Based on statistical analysis of 161 accident cases, the frequency and probability of accidents caused by single-factor coupling, double-factor coupling, and multi-factor coupling are shown in Table 2.

To calculate the coupling value $T$ for each form of interaction among risk factors, the probabilities associated with different risk factor coupling types must first be determined. The calculation formulas for each type of risk coupling probability are as follows:

**Single-factor risk coupling probability.** When the *Individuals* risk factor remains safe (I=0) and the other three risk factors have uncertain states, the accident probability is computed as follows:

$$p_{0\cdots} = p_{0000} + p_{0100} + p_{0010} + p_{0001} + p_{0101} + p_{0110} + p_{0011} + p_{0111}. \tag{6}$$

By analogy, $p_{1\cdots}$, $p_{\cdot 0\cdots}$, ..., $p_{\cdots 1}$ can be computed similarly.

**Double-factor risk coupling probability.** When both *Individuals* and *vehicle* risk factors remain safe (I=0, V=0), and the states of the other two risk factors are uncertain, the accident probability is computed as follows:

$$p_{00\cdots} = p_{0000} + p_{0010} + p_{0001} + p_{0011}. \tag{7}$$

By analogy, $p_{1\cdots}$, $p_{\cdot 0\cdots}$, ..., $p_{\cdots 1}$ can be computed similarly.

**Multi-factor risk coupling probability.** When *Individuals*, *vehicle*, and *environment* risk factors do not occur (i.e., all in state 0), the accident probability is computed as follows:

$$p_{000\cdot} = p_{0000} + p_{0001}. \tag{8}$$

By analogy, $p_{100\cdot}$, $p_{010\cdot}$, ..., $p_{11\cdot 1}$ can be computed similarly, and those results are shown in Table 3.

Subsequently, by applying Formulas (1), (2), and (3), the coupling value $T$ for each risk-factor interaction scenario can be calculated. The results of these calculations are shown in Table 4.

From analysis of the N-K model results, the following observations emerge:

**Table 2. Frequency and probability of different risk coupling forms in road freight transport accidents.**

| Coupling Type | Coupling Factors | Coupling Count | Frequency |
|---|---|---|---|
| 4*Single-Factor Coupling | Individuals (I) | 26 | $P_{1000} = 0.1625$ |
| | Vehicle (V) | 2 | $P_{0100} = 0.0125$ |
| | Environment (E) | 1 | $P_{0010} = 0.0063$ |
| | Management (M) | 0 | $P_{0001} = 0.000$ |
| 5*Two-Factor Coupling | Individuals–Vehicle | 21 | $P_{1100} = 0.1313$ |
| | Individuals–Environment | 14 | $P_{1010} = 0.0813$ |
| | Individuals–Management | 7 | $P_{1001} = 0.0438$ |
| | Vehicle–Environment | 2 | $P_{0110} = 0.0125$ |
| | Vehicle–Management | 3 | $P_{0101} = 0.0188$ |
| | Environment–Management | 1 | $P_{0011} = 0.0063$ |
| 6*Multi-Factor Coupling | I–V–E | 32 | $P_{1110} = 0.2000$ |
| | I–V–M | 11 | $P_{1101} = 0.0688$ |
| | I–E–M | 11 | $P_{1011} = 0.0688$ |
| | V–E–M | 2 | $P_{0111} = 0.0125$ |
| | I–V–E–M | 28 | $P_{1111} = 0.1750$ |

**Table 3. Accident probabilities under different conditions of risk coupling for heavy freight vehicles.**

**Single-Factor Risk Coupling Probability**

| | $p_{0\cdots}$ | $p_{1\cdots}$ | $p_{\cdot0\cdot\cdot}$ | $p_{\cdot1\cdot\cdot}$ | $p_{\cdot\cdot0\cdot}$ | $p_{\cdot\cdot1\cdot}$ | $p_{\cdot\cdot\cdot0}$ | $p_{\cdot\cdot\cdot1}$ |
|---|---|---|---|---|---|---|---|---|
| Value | 0.0682 | 0.9315 | 0.3725 | 0.6272 | 0.4347 | 0.5650 | 0.4595 | 0.5402 |

**Double-Factor Risk Coupling Probability**

| | $p_{00\cdot\cdot}$ | $p_{01\cdot\cdot}$ | $p_{10\cdot\cdot}$ | $p_{11\cdot\cdot}$ | $p_{0\cdot0\cdot}$ | $p_{0\cdot1\cdot}$ | $p_{1\cdot0\cdot}$ | $p_{1\cdot1\cdot}$ |
|---|---|---|---|---|---|---|---|---|
| Value | 0.0124 | 0.0434 | 0.3601 | 0.5714 | 0.0310 | 0.0372 | 0.4037 | 0.5278 |

| | $p_{\cdot00\cdot}$ | $p_{\cdot01\cdot}$ | $p_{\cdot10\cdot}$ | $p_{\cdot11\cdot}$ | $p_{\cdot0\cdot0}$ | $p_{\cdot0\cdot1}$ | $p_{\cdot1\cdot0}$ | $p_{\cdot1\cdot1}$ |
|---|---|---|---|---|---|---|---|---|
| Value | 0.2049 | 0.1676 | 0.2298 | 0.2608 | 0.2111 | 0.1614 | 0.2484 | 0.3788 |

| | $p_{\cdot\cdot00}$ | $p_{\cdot\cdot01}$ | $p_{\cdot\cdot10}$ | $p_{\cdot\cdot11}$ | $p_{0\cdot\cdot0}$ | $p_{0\cdot\cdot1}$ | $p_{1\cdot\cdot0}$ | $p_{1\cdot\cdot1}$ |
|---|---|---|---|---|---|---|---|---|
| Value | 0.2298 | 0.2049 | 0.2297 | 0.3353 | 0.0310 | 0.0372 | 0.4285 | 0.5030 |

**Multi-Factor Risk Coupling Probability**

| | $p_{000\cdot}$ | $p_{100\cdot}$ | $p_{010\cdot}$ | $p_{001\cdot}$ | $p_{110\cdot}$ | $p_{101\cdot}$ | $p_{011\cdot}$ | $p_{111\cdot}$ |
|---|---|---|---|---|---|---|---|---|
| Value | 0 | 0.2049 | 0.0310 | 0.0124 | 0.1988 | 0.1552 | 0.0248 | 0.3726 |

| | $p_{\cdot000}$ | $p_{\cdot100}$ | $p_{\cdot010}$ | $p_{\cdot001}$ | $p_{\cdot110}$ | $p_{\cdot101}$ | $p_{\cdot011}$ | $p_{\cdot111}$ |
|---|---|---|---|---|---|---|---|---|
| Value | 0.1304 | 0.0994 | 0.0807 | 0.0745 | 0.1490 | 0.1304 | 0.0869 | 0.2484 |

| | $p_{0\cdots00}$ | $p_{1\cdots00}$ | $p_{0\cdots10}$ | $p_{0\cdots01}$ | $p_{1\cdots10}$ | $p_{1\cdots01}$ | $p_{0\cdots11}$ | $p_{1\cdots11}$ |
|---|---|---|---|---|---|---|---|---|
| Value | 0.0124 | 0.2174 | 0.0186 | 0.0186 | 0.2111 | 0.1863 | 0.0186 | 0.3167 |

| | $p_{00\cdots0}$ | $p_{10\cdots0}$ | $p_{01\cdots0}$ | $p_{00\cdots1}$ | $p_{11\cdots0}$ | $p_{10\cdots1}$ | $p_{01\cdots1}$ | $p_{11\cdots1}$ |
|---|---|---|---|---|---|---|---|---|
| Value | 0.0124 | 0.2049 | 0.0248 | 0.0062 | 0.2236 | 0.1552 | 0.0310 | 0.3478 |

**Table 4. Coupling values under different risk coupling forms.**

| Coupling Type | Coupling Factors | Risk Coupling Code | Risk Coupling Value |
|---|---|---|---|
| 6*Double-Factor Coupling | Individuals–Vehicle | $T_{21}$ | 0.0624 |
| | Individuals–Environment | $T_{22}$ | 0.0348 |
| | Individuals–Management | $T_{23}$ | 0.0302 |
| | Vehicle–Environment | $T_{24}$ | 0.0054 |
| | Vehicle–Management | $T_{25}$ | 0.0096 |
| | Environment–Management | $T_{26}$ | 0.0182 |
| 5*Multi-Factor Coupling | I–V–E | $T_{31}$ | 0.1597 |
| | I–V–M | $T_{32}$ | 0.0956 |
| | I–E–M | $T_{33}$ | 0.0556 |
| | V–E–M | $T_{34}$ | 0.0287 |
| | I–V–E–M | $T_4$ | 0.2107 |

- **Multi-factor coupling.** As the number of coupled risk factors increases, the coupling value $T$ increases in a gradient fashion [3,5,7]. Specifically, when all four factors—Individuals, vehicle, environment, and management—are coupled, the coupling value can reach as high as 0.2107, indicating that effective control of multi-factor coupling is key to preventing road freight transport accidents.

- **Individuals–vehicle coupling.** The Individuals–vehicle double-factor coupling has a coupling value of 0.0624, which is notably higher than other double-factor coupling forms. Accident statistics suggest that when vehicle malfunctions coincide with unsafe driver behaviors (e.g., speeding, fatigue driving, or illegal overtaking), accidents are likely. Therefore, to avoid Individuals–vehicle double-factor coupling, vehicles must undergo thorough safety checks before departure, and drivers should receive real-time safety reminders through dynamic monitoring systems while on the road.

- **Triple-factor coupling.** Among triple-factor couplings, Individuals–vehicle–environment (0.1597) and Individuals–vehicle–management (0.0956) have substantially higher coupling values than the other forms. This indicates that the interconnection between Individuals factors and vehicle conditions is especially significant in road freight transport. Combined

with the double-factor analysis, if an unfavorable environment and ineffective management coexist alongside Individuals–vehicle coupling, the probability of accidents increases considerably.

## Complex-network metrics

Using Gephi, we generated the risk factor network and calculated the degree, closeness, and betweenness centrality for each risk factor node [12–14]. The results are shown in Table 5.

**Degree centrality.** Degree centrality clearly reflects the influence and position of each risk factor in the network. In general, Individuals-related and vehicle-related factors have higher degree values. Notably, the nodes with the highest degrees—such as overloading/oversized loading (V08), defective braking system (V02), inadequate road safety inspection (M02), failure to observe road conditions (H04), improper driving operations (H06), and adverse weather (E01)—can be considered key risk factors in the network (often referred to as network "hubs"). As a node's degree increases, the coupling effects between that factor and other risk factors tend to intensify.

**Closeness centrality.** Several risk factors show relatively high closeness centrality values, indicating they are situated near the center of the risk factor network. In particular, inadequate road safety inspection (M02), improper driving operations (H06), adverse weather (E01), and speeding (H01) have among the highest closeness centralities. These nodes are very central in the network and thus play a critical role in road freight transport risk propagation; a change in any of these factors can quickly influence many other parts of the system.

**Table 5. Network centrality measures for each risk factor (node) in road freight transport accidents.**

| Risk Factor | Degree | Closeness Centrality | Betweenness Centrality |
|---|---|---|---|
| H01 | 17 | 0.7647 | 9.9481 |
| H02 | 10 | 0.6190 | 1.7917 |
| H03 | 9 | 0.6047 | 1.7853 |
| H04 | 18 | 0.6431 | 9.5781 |
| H05 | 5 | 0.5532 | 0.0833 |
| H06 | 18 | 0.7647 | 9.3981 |
| H07 | 17 | 0.7429 | 7.5576 |
| H08 | 8 | 0.5909 | 0.8357 |
| H09 | 16 | 0.7027 | 11.3361 |
| V01 | 13 | 0.6667 | 3.3189 |
| V02 | 19 | 0.7879 | 10.1584 |
| V03 | 10 | 0.6190 | 1.7500 |
| V04 | 14 | 0.6842 | 3.5460 |
| V05 | 16 | 0.6551 | 6.1766 |
| V06 | 11 | 0.6341 | 1.6512 |
| V07 | 16 | 0.7222 | 5.6302 |
| V08 | 20 | 0.8125 | 14.8168 |
| E01 | 18 | 0.7647 | 9.0691 |
| E02 | 10 | 0.6190 | 1.8262 |
| E03 | 11 | 0.6190 | 1.6305 |
| E04 | 12 | 0.6500 | 3.1182 |
| E05 | 13 | 0.6667 | 3.3290 |
| E06 | 15 | 0.7027 | 6.3330 |
| E07 | 16 | 0.7222 | 6.4867 |
| E08 | 16 | 0.7222 | 8.0391 |
| M01 | 16 | 0.7222 | 7.2630 |
| M02 | 18 | 0.7647 | 14.3299 |

**Betweenness centrality.** A few risk factors act as important bridges in the network, as evidenced by high betweenness centrality. In this network, inadequate road safety inspection (M02), illegal lane changing (H09), defective braking system (V02), and speeding (H01) appear to be effective "bridge" nodes. By managing and controlling these factors, it is possible to break critical connections in the risk network, thereby reducing the likelihood of road freight transport accidents.

## Discussion

### Potential risk factor coupling analysis

To integrate the risk-coupling insights from the N-K model with the complex network results, we began by identifying which first-level categories (I, V, E, M) each specific risk factor could potentially involve. For instance, a driver's speeding behavior (H01) could couple with defective vehicle brakes (V02) and adverse weather (E01). Table 6 enumerates each factor, its corresponding first-level category, and all possible coupling forms (single, double, triple, or quadruple) in which that factor might appear.

Recognizing a factor's potential coupling forms enables referencing the coupling probability values ($T$) from the N-K model (see Sect 5.1) to assign higher or lower importance to that factor in the complex network. Specifically, if a factor frequently participates in high-$T$ coupling scenarios—e.g., involving multiple domains—it exerts a stronger influence on overall

**Table 6. Nodes, corresponding first-level risk factors, and potential coupling forms.**

| Risk Factor | Individuals | Vehicle | Environment | Management | Potential Coupling Form |
|---|---|---|---|---|---|
| H01 | 1 | 1 | 1 | 0 | I–V–E |
| H02 | 1 | 1 | 1 | 1 | I–V–E–M |
| H03 | 1 | 1 | 1 | 0 | I–E–M |
| H04 | 1 | 1 | 1 | 1 | I–V–E–M |
| H05 | 1 | 0 | 1 | 1 | I–E–M |
| H06 | 1 | 1 | 1 | 1 | I–V–E–M |
| H07 | 1 | 1 | 1 | 1 | I–V–E–M |
| H08 | 1 | 1 | 1 | 1 | I–V–E–M |
| H09 | 1 | 1 | 1 | 0 | I–V–E |
| V01 | 1 | 1 | 1 | 1 | I–V–E–M |
| V02 | 1 | 1 | 1 | 1 | I–V–E–M |
| V03 | 1 | 1 | 1 | 0 | I–V–E |
| V04 | 1 | 1 | 1 | 0 | I–V–E |
| V05 | 1 | 1 | 1 | 1 | I–V–E–M |
| V06 | 0 | 1 | 0 | 0 | V |
| V07 | 1 | 1 | 1 | 1 | I–V–E–M |
| V08 | 1 | 1 | 1 | 1 | I–V–E–M |
| E01 | 1 | 1 | 1 | 1 | I–V–E–M |
| E02 | 1 | 1 | 1 | 0 | I–V–E |
| E03 | 1 | 1 | 1 | 1 | I–V–E–M |
| E04 | 1 | 1 | 1 | 1 | I–V–E–M |
| E05 | 1 | 1 | 1 | 1 | I–V–E–M |
| E06 | 1 | 1 | 1 | 1 | I–V–E–M |
| E07 | 1 | 1 | 1 | 1 | I–V–E–M |
| E08 | 1 | 0 | 1 | 1 | I–E–M |
| M01 | 1 | 1 | 1 | 1 | I–V–E–M |
| M02 | 1 | 1 | 1 | 1 | I–V–E–M |

accident probability. Conversely, if a factor tends to appear in lower-probability couplings, its relative importance is lower.

Using this approach, we revised the centrality results of the complex network by weighting each factor according to its relevant coupling probabilities. Figs 3 and 4 compare the "pre-revision" and "post-revision" radar charts of degree, closeness, and betweenness centralities. A visual inspection reveals notable shifts in the prominence of certain risk factors once coupling is taken into account. Broadly, factors that appear in high-probability coupling combinations (especially those linking Individuals and vehicle domains) became more pronounced after revision.

**Key findings.** Overall, the revised charts underscore how multi-factor interactions amplify accident risk. Our N-K model confirms that coupling across domains (particularly Individuals–vehicle–environment or the four-way coupling I–V–E–M) significantly increases crash likelihood. The complex network analysis reveals specific "hub" and "bridge" nodes,

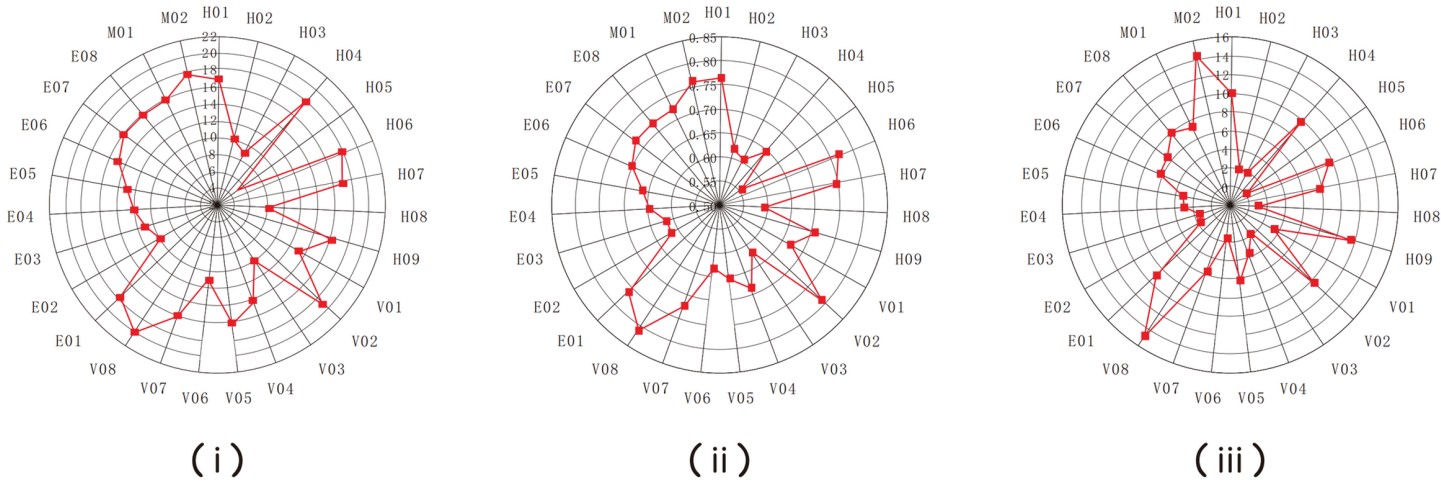

**Fig 3. Pre-revision radar charts of (i) degree, (ii) degree centrality, and (iii) closeness centrality for all risk factors.**

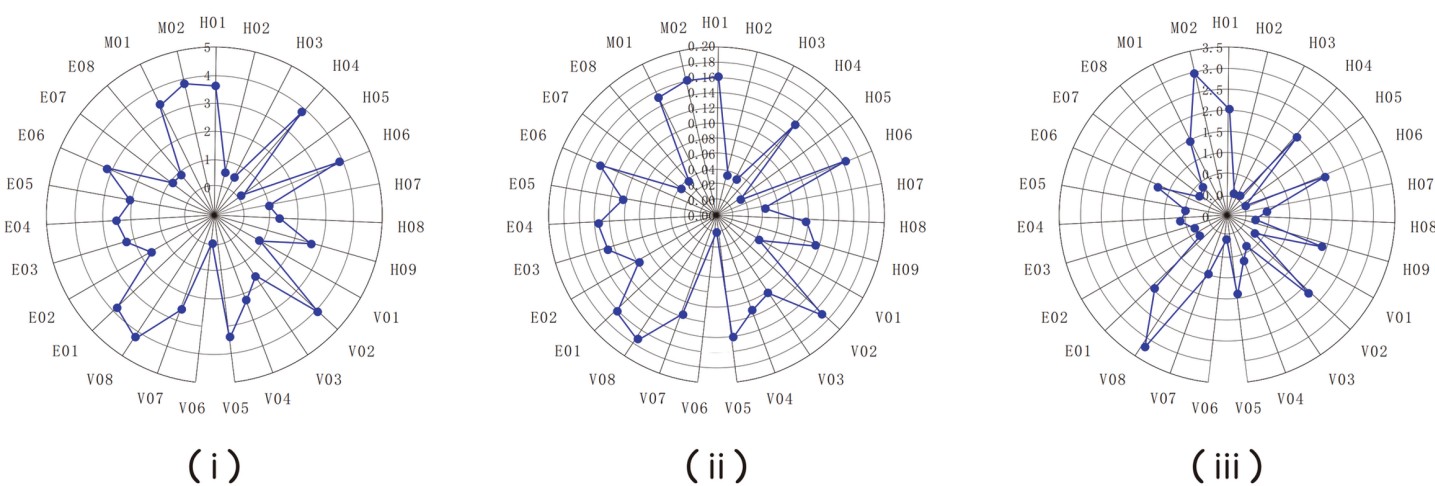

**Fig 4. Post-revision radar charts of (i) degree, (ii) degree centrality, and (iii) closeness centrality after adjusting for coupling values.**

such as overloading (V08), defective braking systems (V02), inadequate road inspection (M02), speeding (H01), and failure to observe road conditions (H04). The high degrees of these nodes mean they co-occur frequently with other factors, while their high betweenness values indicate that controlling these factors could disrupt many risk transmission paths in the network.

By combining the N-K model's quantitative coupling values with the network's topological insights, we obtain a more holistic picture of road freight transport safety. In particular, the results highlight that:

1. **Vehicle–Individual Coupling is Paramount:**Overloading, brake defects, and unsafe driving behaviors (e.g., speeding) frequently co-occur, suggesting that targeted monitoring and strict enforcement against these combined violations would likely yield significant safety benefits.
2. **Environmental Challenges Interact with Other Factors:**Adverse weather (E01) and slippery road surfaces (E06) notably elevate risk when paired with suboptimal driving or vehicle conditions.
3. **Management Gaps Amplify Everything Else:**Inadequate road safety inspection (M02) or insufficient vehicle dynamic monitoring (M01) allow minor single-factor issues to escalate into major accidents. Improved oversight measures can thus mitigate risk across all domains.

In summary, the post-revision charts and the integrated analysis indicate that focusing resources on a small subset of high-coupling, high-centrality risk factors can yield outsized improvements in safety. The following sections propose targeted measures and discuss broader implications for policy and industry practice.

## Preventive measures aligned with key risk factors

From the combined N-K model and complex network results, several risk factors appear consistently in high-risk couplings or occupy critical positions in the network. This section proposes targeted preventive measures for each category of factors—Individuals, vehicle, environment, and management—highlighting both the rationale and potential implementation strategies.

**Individuals factors.** Key Individuals-related risks that frequently contribute to road freight accidents include speeding (H01), failure to observe road conditions (H04), and improper driving operations (H06). These behaviors stem from inadequate safety awareness, driver fatigue, and insufficient regulatory compliance [1,2,23,27,28].

Recommended Measures:

1. *Strengthen and Diversify Driver Training.* Require periodic refresher courses that integrate defensive driving modules, speed management, and hazard perception. Before license renewals, implement stricter theoretical and on-road assessments to ensure only qualified drivers operate cargo vehicles.
2. *Implement Fatigue Management Programs.* Introduce mandatory rest schedules with electronic logging of driving hours. Use real-time camera-based detection or alertness trackers to reduce fatigue-related errors, particularly on long-haul trips.
3. *Introduce Defensive Driving Courses.* Encourage or mandate training in defensive driving, emphasizing proper lane-keeping, maintaining safe following distances, and rapid hazard detection. Such courses could be integrated into carrier onboarding programs and annual safety evaluations.

**Vehicle factors.** The vehicle-related factors most closely tied to accidents are defective braking systems (V02) and overloading or oversized loading (V08). Both issues significantly increase stopping distance and compromise vehicle stability during emergencies [29–32].
Recommended Measures:

1. *Rigorous Inspection and Maintenance Programs.* Require logistics companies to conduct systematic pre- and post-trip inspections, focusing on brakes, tires, and steering. Enforce a "no dispatch" policy if any critical defect is identified.
2. *Stricter Enforcement of Loading Limits.* Reinforce weigh station operations to penalize overloading, and integrate weigh-in-motion (WIM) sensors on major freight routes for continuous monitoring without halting traffic flow.
3. *Accountability Mechanisms for Cargo Owners.* Assign legal liability to cargo owners or brokers who repeatedly permit or encourage illegal loading. For carriers with multiple violations, consider suspending licenses or revoking operating permits.

**Environment-related factors.** Environmental conditions, especially adverse weather (E01), amplify risks posed by minor driving errors. Snow, fog, and heavy rain reduce visibility and road friction, turning routine maneuvers into potential accidents [24–26,33].
Recommended Measures:

1. *Coordinated Meteorological Data and Road Safety Management.* Develop real-time data-sharing platforms between meteorological agencies and highway operators to ensure swift dissemination of severe weather alerts, along with dynamic traffic control measures (e.g., temporary speed limits, convoy driving).
2. *Upgraded Infrastructure.* Improve road surfaces with higher-friction materials and install visible signage in high-risk zones (fog-prone or heavy-snowfall areas). Deploy automated variable message signs (VMS) to alert drivers about sudden weather changes.
3. *Robust Emergency Preparedness.* Position emergency vehicles and crews strategically along high-traffic corridors, ensuring rapid response. Maintain well-stocked equipment for rescue and road clearance to minimize secondary collisions under poor weather conditions.

**Management factors.** Among management-related risks, inadequate road safety inspection (M02) appears more pervasive than insufficient vehicle dynamic monitoring (M01). Both significantly influence safety outcomes by either permitting infrastructure hazards to persist or allowing unsafe driving behaviors to go unaddressed [8,17,21,23].
Recommended Measures:

1. *Comprehensive Road and Cargo Oversight.* Conduct regular audits of highways, focusing on guardrails, lighting, and signage. Likewise, enforce spot checks on cargo loading sites, ensuring load stability and preventing unsecured or shifting cargo.
2. *Enhanced Dynamic Monitoring Systems.* Require carriers to maintain active GPS and telematics devices throughout each trip, flagging speeding, abrupt braking, or evasive maneuvers in real time. Promptly repair any non-functional monitoring device to avoid oversight gaps.
3. *Clear Accountability and Data Management.* Assign explicit responsibilities for data collection and analysis within each freight company. Regulatory agencies should periodically review logs to detect and address recurring violations or neglected system alerts.

These recommendations focus on high-impact measures that directly target the most critical risk factors identified via the N-K model and complex network analysis. A concerted effort—combining driver education, rigorous vehicle maintenance, adaptive road management, and robust oversight mechanisms—can disrupt hazardous couplings that frequently lead to severe accidents in road freight transport.

## Limitations and future research directions

**Limitations.** Although this research integrates the N-K model with a complex network approach using official accident investigation reports, several limitations remain. First, the quality and completeness of official reports can vary significantly, potentially causing under-reporting or uneven emphasis on contributory factors. Due to these inconsistencies, we initially identified only 203 reports sufficiently detailed for in-depth analysis, of which 160 were ultimately confirmed and selected after consultation with local police. This represents a relatively small sample size compared to the vast scale of China's road freight transport sector. Second, the reliance on manual review of lengthy accident reports limited our ability to accurately quantify the frequency of each risk factor. Consequently, we had to rely primarily on the factor significance levels as officially documented, rather than precise frequency-based weighting. Third, both the N-K model and the complex network utilized in this study provide only static snapshots of risk factors. Therefore, they do not capture dynamic variations in the importance or interactions of these factors, which could occur due to seasonal fluctuations or changes in traffic regulations

**Future research directions.** Given these limitations, future research could leverage natural language processing (NLP) and automated text-mining techniques to efficiently process a substantially larger volume of accident reports. Employing NLP would enable precise frequency-based weighting of risk factors by accurately capturing their occurrence across numerous reports, enhancing the robustness and representativeness of analytical findings. Additionally, future studies should consider adopting dynamic or longitudinal modeling frameworks that account for temporal shifts in risk factor importance and interactions, especially in response to seasonal variations and regulatory updates. Integration with advanced AI tools, such as machine learning for predictive driver behavior analysis or real-time monitoring systems, could further refine the predictive accuracy and practical applicability of risk assessments.

## Conclusions

In summary, this study combined an N-K model with a complex network framework to analyze coupling effects among 27 risk factors in road freight transport accidents. The findings reveal that multi-factor interactions—particularly those involving both individual and vehicle domains—strongly amplify accident probabilities. Additionally, complex network metrics identify overloading, defective braking systems, and inadequate road safety inspection as "hub" or "bridge" factors that substantially drive risk propagation. Managing these specific factors through stricter enforcement, comprehensive vehicle checks, and driver training can disrupt key points of risk accumulation. Overall, the methodology and results presented here offer a robust foundation for targeted policy interventions and pave the way for future research exploring dynamic and AI-enhanced modeling of transportation safety.

## Supporting information

**S1 Data. Data sources for figures and tables.**
(ZIP)

## Acknowledgments

We are grateful to the anonymous reviewers for their insightful comments and constructive suggestions. We also thank the Zhejiang Provincial Department of Culture, Radio, Television and Tourism and the Zhejiang Provincial Science and Technology Department for the support of this research.

## Author contributions

**Conceptualization:** Runhua Huang.

**Data curation:** Runhua Huang.

**Formal analysis:** Runhua Huang.

**Funding acquisition:** Runhua Huang.

**Validation:** Huichao Guo.

**Visualization:** Huichao Guo.

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
