## [Decision Letter · Decision Letter 0]

3 Jul 2025

PONE-D-25-10686Coupling Analysis of Risk Factors in Road Cargo Transport Accidents and Preventive Measures with N-K ModelPLOS ONE

Dear Dr. Huang,

Thank you for submitting your manuscript to PLOS ONE. After careful consideration, we feel that it has merit but does not fully meet PLOS ONE’s publication criteria as it currently stands. Therefore, we invite you to submit a revised version of the manuscript that addresses the points raised during the review process.

We look forward to receiving your revised manuscript.

Kind regards,

Mohammad Hossein Ebrahimi

Academic Editor

PLOS ONE

‘This research was funded by the Zhejiang Provincial Department of Culture, Radio, Television and Tourism, grant number 2024KYZ009.”

Reviewers' comments:

Reviewer's Responses to Questions

**Comments to the Author**

1. Is the manuscript technically sound, and do the data support the conclusions?

Reviewer #1: Partly

Reviewer #2: Yes

2. Has the statistical analysis been performed appropriately and rigorously? 

Reviewer #1: I Don't Know

Reviewer #2: I Don't Know

3. Have the authors made all data underlying the findings in their manuscript fully available?

Reviewer #1: Yes

Reviewer #2: Yes

4. Is the manuscript presented in an intelligible fashion and written in standard English?

Reviewer #1: Yes

Reviewer #2: Yes

5. Review Comments to the Author

Reviewer #1: - In the abstract section, the conclusion part should be rewritten based on the main findings of the study.

- Many of the statements presented in the introduction lack thematic relevance to the study objectives. The introduction should primarily focus on the key factors influencing cargo transport accidents. Subsequently, the importance of investigating these factors and analyzing their interrelationships should be clearly articulated to help readers understand the study's main line. The sentences should then elaborate on methodological approaches for examining these relationships, highlighting the advantages of the N-K model over alternative models such as BN that led to its selection by the authors. In its current form, the introduction provides only a superficial treatment of the subject and requires substantial strengthening and thorough revision.

- How 27 specific risk factors were identified accident reports? Are these documented as root causes or direct causes in the official accident reporting documentation? As we know, root causes differ from contributory factors.

- The discussion section should go beyond merely restating the results. Authors must critically analyze and interpret the findings in depth, contextualizing them within existing literature. A superficial repetition of results is insufficient. Additionally, incorporating study limitations, recommendations for future work, practical implications, and a clear conclusion would significantly strengthen this section.

Reviewer #2: An interesting study was conducted and well written. Congratulations to the researchers. Only two points need further explanation, which are suggested to be included.

Instead of using "human factors", use "individual factors". Human factors cover a wider range.

It is not clear how risk factors are extracted from reports.

Is it possible that a risk factor has been repeated many times in different accidents? How are the more important risk factors weighted in the model?

The body of the manuscript should follow the journal format.

6. PLOS authors have the option to publish the peer review history of their article (what does this mean?). If published, this will include your full peer review and any attached files.

Reviewer #1: **Yes: **Leila Omidi

Reviewer #2: **Yes: **Payam khanlari

---

## [Author Response · Author response to Decision Letter 1]

13 May 2025

To The Editor,

PLOS ONE.

9th May 2024.

Dear Sir/Madam,

RE: RESPONSE TO REVIEWER’S COMMENTS: MANUSCRIPT NO: PONE-D-25-10686

We are grateful to you and the reviewers for the thorough evaluation of our manuscript. We greatly appreciate the time and constructive feedback offered, and we have carefully addressed all comments. Below, we provide point-by-point responses to each of the reviewers’ suggestions and explain the corresponding revisions made in our manuscript (as reflected in the revised version). The responses are highlighted in blue.

Revision format Issue

1. We note that your manuscript is not formatted using one of PLOS ONE’s accepted file types. Please reattach your manuscript as one of the following file types: .doc, .docx, .rtf, or .tex (accompanied by a .pdf).

If your submission was prepared in LaTex, please submit your manuscript file in PDF format and attach your .tex file as “other.”

RE: That has been addressed, my submission was prepared in Latex, and now I have attach my text file as “other” in the new revision.

2. Please ensure that you refer to Figure 3 & 4 in your text as, if accepted, production will need this reference to link the reader to the figure.

RE: Figures 3 and 4 are refered in the text, and please check line 281.

3. Please ensure that you refer to Table 3 & 4 in your text as, if accepted, production will need this reference to link the reader to the Table.

RE: That has been addressed, tables 3 and 4 are refered in the text, and please check line 215 for table 3 and line 218 for table 4.

and

RE: That is corrected. We have now followed the PLOS ONE template available at https://journals.plos.org/plosone/s/latex .

2.Please note that PLOS ONE has specific guidelines on code sharing for submissions in which author-generated code underpins the findings in the manuscript. In these cases, we expect all author-generated code to be made available without restrictions upon publication of the work. Please review our guidelines at https://journals.plos.org/plosone/s/materials-and-software-sharing#loc-sharing-code and ensure that your code is shared in a way that follows best practice and facilitates reproducibility and reuse.

RE: All numerical results presented in the manuscript—including single-, double-, and multi-factor coupling values (Equations 1–3) and the network centrality measures (degree, closeness, betweenness Equations 4–5)—were obtained by straightforward evaluation of the mathematical expressions printed in the text. Each formula operates on the 27 risk-factor indicators in the supplied “Source data” spreadsheet and can be replicated using any standard data-analysis tool with simple operations—frequency counts, base-2 logarithms, and shortest-path summations—which can be carried out in common environments such as R, MATLAB, Python, or even Microsoft Excel. Any researcher with access to the data and the formulas in the Materials and methods section of the manuscript can regenerate our tables in just a few steps. Therefore, there are no specific code for calculation provided for this paper.

‘This research was funded by the Zhejiang Provincial Department of Culture, Radio, Television and Tourism, grant number 2024KYZ009.” Please state what role the funders took in the study. If the funders had no role, please state: "The funders had no role in study design, data collection and analysis, decision to publish, or preparation of the manuscript."If this statement is not correct you must amend it as needed. Please include this amended Role of Funder statement in your cover letter; we will change the online submission form on your behalf.

RE: Here is the updated funding information for you to kindly upload it in the system:

The authors gratefully acknowledge the financial support for this research by the following organizations and agencies: The Soft Science Research Program of Zhejiang Province, 429 Grant Number: 2025C35083 and Zhejiang Provincial Department of Culture, Radio, Television and Tourism, grant number 2024KYZ009. The funders had no role in study design, data collection and analysis, decision to publish, or preparation of the manuscript.

RE: Data Availability: Yes - all data are fully available without restriction right now. All data files have been uploaded to GitHub and you can access that with the direct link: https://github.com/RunhuaHuang/Data-Pone

5.Please include your full ethics statement in the ‘Methods’ section of your manuscript file. In your statement, please include the full name of the IRB or ethics committee who approved or waived your study, as well as whether or not you obtained informed written or verbal consent. If consent was waived for your study, please include this information in your statement as well.

RE: That has been addressed, and please check Page 3, Line 91-95.

Yours faithfully,

Runhua Huang

The Chinese University of Hong Kong, Shenzhen

Runhuahuang@link.cuhk.edu.cn

BELOW IS OUR RESPONSE TO THE REVIEWER’S COMMENTS IN BLUE

Reviewer #1

1)In the abstract section, the conclusion part should be rewritten based on the main findings of the study.

RE: We thank the reviewer for this helpful suggestion. In the revised manuscript, we have substantially rewritten the conclusion in the abstract to succinctly reflect the primary findings and their significance. Specifically, we:

1.Clarified which risk factors consistently showed high coupling strength in our N-K model and complex network analysis.

2.Summarized how these results inform targeted preventive measures for road freight accidents.

These updates ensure that the abstract’s conclusion now aligns with the core evidence and insights gained from the study.

2)Many of the statements presented in the introduction lack thematic relevance to the study objectives. The introduction should primarily focus on the key factors influencing cargo transport accidents. Subsequently, the importance of investigating these factors and analyzing their interrelationships should be clearly articulated to help readers understand the study's main line. The sentences should then elaborate on methodological approaches for examining these relationships, highlighting the advantages of the N-K model over alternative models such as BN that led to its selection by the authors. In its current form, the introduction provides only a superficial treatment of the subject and requires substantial strengthening and thorough revision.

RE: This has been addressed. In the revised version we have:

1.Reorganized the introduction to establish a clearer logical flow, prioritizing the principal risk factors in road cargo transport accidents (individual, vehicle, environment, management) and providing more contextual references.

2.Added new text explaining the importance of systematically examining the interdependencies among these risk factors. This ensures readers immediately grasp the motivation for exploring risk-factor coupling.

3.Expanded our methodological justification for choosing the N-K model over other approaches such as Bayesian Networks (BN). Specifically, we explain how the N-K model offers direct advantages for assessing how multi-factor interactions contribute to emergent accident probability, while BN approaches, although powerful, can be more cumbersome for quantifying higher-order factor couplings.

3)How 27 specific risk factors were identified accident reports? Are these documented as root causes or direct causes in the official accident reporting documentation? As we know, root causes differ from contributory factors.

RE: We sincerely appreciate this request for clarification, as the factor identification procedure is central to our study. In Section 2, we have expanded our explanation of how we derived the final set of 27 risk factors. More specifically:

1.Initial Review: Each accident investigation report was read in full to identify any mention of causal factors. This encompassed explicit “root causes,” “direct causes,” and “contributory factors.” Factors described with synonymous terms (e.g., “excessive speed” vs. “overspeeding”) were consolidated under a single representative label (e.g., “speeding”).

2.Coding and Consolidation: We coded each reported cause or factor (e.g., “driver speeding,” “vehicle brake failure,” “poor road design”) and consolidated synonymous or repeated items. For instance, “excessive speed” and “overspeeding” were grouped together as “speeding.”

3.Local Consultation and Confirmation: For each accident, we attempted to verify our coding with the local traffic police who had overseen the investigation. In 160 of the 203 reports, the police confirmed or clarified our factor classifications. The remaining 43 reports were excluded from subsequent analyses due to disagreements about cause attribution, difficulty contacting the relevant officers, or insufficient cooperation. Additionally, factors that were rarely reported or considered tangential by both the investigation reports and the police were merged with more prevalent factors or removed entirely.

4.Categorization: We then mapped the finalized factors to four top-level categories based on the “I–V–E–M” framework: individual, vehicle, environment, and management. This final set served as the basis for subsequent N-K model and complex network analyses.

5.Root vs. Contributory Causes: We collected and examined every “root cause” and “contributory cause” mentioned in the accident reports; for clarity in our dataset, each accident’s root cause was highlighted in red. However, our primary objective was to assess how various causes—regardless of their designated level of importance—might couple or interact to exacerbate accident risks. Consequently, we did not segment causes by “root” vs. “contributory” classification in our final analysis. Instead, our focus remained on the coupling relationships among all identified factors, allowing us to capture the synergistic effects that different causes may collectively exert on accident likelihood.

By detailing these procedures, we aim to underscore the rigor of our factor identification process and to clarify that both “root causes” and “contributory factors” are included so as to provide a complete representation of all relevant risk elements. This inclusive approach supports a more holistic assessment of factor interplay, which is central to our N-K model and complex network analyses.

4)The discussion section should go beyond merely restating the results. Authors must critically analyze and interpret the findings in depth, contextualizing them within existing literature. A superficial repetition of results is insufficient. Additionally, incorporating study limitations, recommendations for future work, practical implications, and a clear conclusion would significantly strengthen this section.

RE: We appreciate the reviewer’s emphasis on enhancing the depth and structure of our Discussion section. In the revised manuscript, we have made the following changes:

1.Section Potential Risk Factor Coupling Analysis and Key Findings Subsection: We provide a more detailed interpretation of our N-K model and complex network results, examining how various factors couple (e.g., Individuals and vehicle) to amplify accident risk. This discussion goes beyond a simple restatement of the findings by highlighting the implications of specific coupling forms and identifying which factors disproportionately elevate overall accident probability.

2.Section Preventive Measures Aligned with Key Risk Factors: Drawing on the critical insights from Section 5.1, we propose more targeted, concrete measures for mitigating high-impact risks. By directly linking these recommendations to the most influential couplings, we aim to offer practical steps that industry practitioners and policymakers can implement.

3.Section Limitations and Future Research Directions: We have introduced a section(Limitations)outlining the boundaries of our current study—for instance, the static nature of the N-K model and the complexities inherent in analyzing accident reports of varying completeness. We also add a section (Future Research Directions) propose potential directions for future research, such as investigating larger-scale datasets, incorporating real-time monitoring or longitudinal data, and integrating advanced modeling techniques (e.g., machine learning) to capture dynamic changes in risk factors over time.

4.Section Conclusions: Finally, we add a concluding section with an integrated summary that reiterates the importance of multi-factor coupling in road cargo transport accidents and underscores how our results can guide both immediate safety interventions and more nuanced research on factor interactions.

Reviewer #2

1)An interesting study was conducted and well written. Congratulations to the researchers. Only two points need further explanation, which are suggested to be included.

RE: Thanks a lot

2)Instead of using "human factors", use "individual factors". Human factors cover a wider range.

RE: We thank the reviewer for suggesting the shift from “human factors” to “individual factors.” We agree that “individual factors” is more precise. Accordingly, we have replaced all references to “human factors” with “individual factors” throughout the manuscript. The revised text consistently uses “Individuals” (I) instead of “Human” (H) in the 4M (I–V–E–M) framework.

3)It is not clear how risk factors are extracted from reports.

RE: In the revised manuscript (Section 2), we have added a more comprehensive description of how risk factors were extracted, which involves four stages: Initial Review, Coding and Consolidation, Local Consultation and Confirmation, and Categorization, please check Line 65-84.

3)Is it possible that a risk factor has been repeated many times in different accidents? How are the more important risk factors weighted in the model?

RE: Yes, a single risk factor can appear multiple times within one accident report. Moreover, multiple factors within the same “4M” category can co-occur in a single accident; for example, a driver might be both fatigued (I02) and speeding (I01).

Although each accident report may distinguish between “root” and “contributory” causes, our primary objective is to investigate how these factors collectively couple to exacerbate accident risk. We therefore do not segment factors by their priority level when constructing the N-K model or complex network. Instead, we focus on the interaction patterns among all identified factors, regardless of their initially assigned status in official reports. This approach allows us to capture the full spectrum of potential factor combinations that could contribute to an accident, without imposing predefined weights or rankings on the factors themselves.

Conclusion and Final Remarks

We sincerely appreciate t

---

## [Decision Letter · Decision Letter 1]

26 Jun 2025

Coupling Analysis of Risk Factors in Road Cargo Transport Accidents and Preventive Measures with N-K Model

PONE-D-25-10686R1

Dear Dr. Huang,

We’re pleased to inform you that your manuscript has been judged scientifically suitable for publication and will be formally accepted for publication once it meets all outstanding technical requirements.

Kind regards,

Mohammad Hossein Ebrahimi

Academic Editor

PLOS ONE

Additional Editor Comments (optional):

Reviewers' comments:

Reviewer's Responses to Questions

**Comments to the Author**

1. If the authors have adequately addressed your comments raised in a previous round of review and you feel that this manuscript is now acceptable for publication, you may indicate that here to bypass the “Comments to the Author” section, enter your conflict of interest statement in the “Confidential to Editor” section, and submit your "Accept" recommendation.

Reviewer #1: (No Response)

Reviewer #2: All comments have been addressed

2. Is the manuscript technically sound, and do the data support the conclusions?

Reviewer #1: Partly

Reviewer #2: Yes

3. Has the statistical analysis been performed appropriately and rigorously? 

Reviewer #1: I Don't Know

Reviewer #2: I Don't Know

4. Have the authors made all data underlying the findings in their manuscript fully available?

Reviewer #1: No

Reviewer #2: Yes

5. Is the manuscript presented in an intelligible fashion and written in standard English?

Reviewer #1: Yes

Reviewer #2: Yes

6. Review Comments to the Author

Reviewer #1: - The quality of the paper has significantly improved after editing; however, two points still need to be addressed:

1- The 4M model should be removed from the introduction section.

2- In the discussion section, the content is treated superficially, and a thorough interpretation of the findings is lacking. First, the most significant findings should be presented, followed by their interpretation and comparison with those of other studies. In its current form, the focus is primarily on proposing control measures, while the comparison of findings with other studies and their in-depth interpretation remain underdeveloped. Given the volume of published literature, this section could be substantially enriched.

Reviewer #2: The manuscript has improved considerably and the authors have addressed the comments well. I would also suggest that a statistician review the data analysis section.

7. PLOS authors have the option to publish the peer review history of their article (what does this mean?). If published, this will include your full peer review and any attached files.

Reviewer #1: No

Reviewer #2: No

---

## [Editor Report · Acceptance letter]

PONE-D-25-10686R1

PLOS ONE

Dear Dr. Huang,

I'm pleased to inform you that your manuscript has been deemed suitable for publication in PLOS ONE. Congratulations! Your manuscript is now being handed over to our production team.

Kind regards,

on behalf of

Dr. Mohammad Hossein Ebrahimi

Academic Editor

PLOS ONE